# Trabectedin Induces Synthetic Lethality via the p53-Dependent Apoptotic Pathway in Ovarian Cancer Cells Without BRCA Mutations When Used in Combination with Niraparib

**DOI:** 10.3390/ijms26072921

**Published:** 2025-03-24

**Authors:** Bongkyun Kang, Sun-Jae Lee, Ki Ho Seol, Yoon Young Jeong, Jung-Hye Choi, Bo-Hyun Choi, Jung Min Ryu, Youn Seok Choi

**Affiliations:** 1Department of Chemistry, College of Natural Science, Kyungpook National University, Daegu 41944, Republic of Korea; xmas84@gmail.com; 2Department of Pathology, School of Medicine, Daegu Catholic University, Daegu 42472, Republic of Korea; pathosjlee@cu.ac.kr; 3Department of Radiation Oncology, School of Medicine, Daegu Catholic University, Daegu 42472, Republic of Korea; khseol@cu.ac.kr; 4Department of Obstetrics and Gynecology, School of Medicine, Daegu Catholic University, Daegu 42472, Republic of Korea; nning@cu.ac.kr; 5Department of Biomedical and Pharmaceutical Sciences, Kyung Hee University, Seoul 02447, Republic of Korea; jchoi@khu.ac.kr; 6Department of Pharmacology, School of Medicine, Daegu Catholic University, Daegu 42472, Republic of Korea; bchoi@cu.ac.kr

**Keywords:** ovarian carcinoma, poly (ADP-ribose) polymerase inhibitor, trabectedin, synthetic lethality, homologous recombination deficiency

## Abstract

This study investigated whether combining niraparib and trabectedin in BRCA-proficient epithelial ovarian cancer induces deficiencies in ssDNA break repair and dsDNA homologous recombination, leading to synthetic lethality. A2780 and SKOV3 ovarian cancer cell lines were treated with niraparib and trabectedin. Cell viability was assessed using CCK-8 assays, while RT-qPCR and Western blot analyzed the expression of DNA repair and apoptosis-related genes. Apoptosis was evaluated via Annexin V/PI assays. The combination therapy exhibited a synergistic effect on A2780 cells but not on SKOV3 cells. Treatment reduced BRCA1, BRCA2, RAD51, PARP1, and PARP2 expression, indicating impaired DNA repair. γ-H2AX levels increased, suggesting DNA damage. The therapy also upregulated p53, PUMA, NOXA, BAX, BAK, and p21, promoting p53-mediated apoptosis and cell cycle arrest. Apoptosis induction was confirmed via Annexin V/PI assays. Silencing p53 with siRNA abolished all synergistic effects in A2780 cells. Niraparib and trabectedin combination therapy impairs DNA repair in BRCA-proficient ovarian cancer, leading to synthetic lethality through p53-dependent apoptosis.

## 1. Introduction

Epithelial ovarian cancer is a disease with a poor prognosis, often detected at an advanced stage due to the lack of early symptoms and adequate screening tests, with high recurrence and mortality rates. According to 2020 statistics, ovarian cancer kills more than 200,000 people a year worldwide [1], and epithelial ovarian cancer is the leading cause of death from gynecologic cancer in developed countries [2]. Epithelial ovarian cancer is classified into low-grade serous carcinoma, high-grade serous carcinoma, mucinous carcinoma, endometrioid carcinoma, clear cell carcinoma, malignant Brenner tumor, mesonephric-like adenocarcinoma, undifferentiated and dedifferentiated carcinoma, carcinosarcoma, and mixed carcinoma according to the WHO classification, of which high-grade serous carcinoma is the most common pathologic type. The standard treatment for newly diagnosed advanced epithelial ovarian cancer is cytoreductive surgery followed by platinum-taxane combination chemotherapy with/without Bevacizumab (anti-VEGF monoclonal antibody), but despite this treatment, the disease has a high recurrence rate and a high mortality rate.

BRCA1 and BRCA2 play an important role in the repair of double-stranded DNA (dsDNA) breaks through homologous recombination, and women with *BRCA1* or *BRCA2* gene mutations are predisposed to cancers such as breast and ovarian cancer. The term ‘BRCAness’ refers to the phenotypic traits with a homologous recombination deficiency (HRD) similar to cells carrying *BRCA1* or *BRCA2* mutations. Poly(ADP-ribose) polymerase (PARP) is an enzyme involved in base excision repair, a key pathway in the repair of single-strand DNA (ssDNA) breaks. PARP inhibitors are agents that inhibit the PARP enzyme and have a pharmacologic mechanism of action that induces cell death through synthetic lethality in cancer cells with HRD such as *BRCA* mutations. Synthetic lethality is a phenomenon where a single genetic event is tolerable for cell survival, but the simultaneous occurrence of multiple genetic events leads to cell death [3]. PARP inhibitors are the first clinically approved drugs designed to exploit synthetic lethality. When the repair of ssDNA breaks is inhibited by PARP inhibitors, ssDNA breaks lead to the occurrence of dsDNA breaks. In the case of cancer cells with HRD, dsDNA break repair involves error-prone DNA repair mechanisms such as Non-Homologous End Joining. Consequently, the accumulation of numerous mutations induces cell death through synthetic lethality.

The genetic concept of synthetic lethality has now been clinically validated through the demonstrated efficacy of PARP inhibitors for the treatment of cancers in patients with HRD. PARP inhibitors, such as olaparib, niraparib, rucaparib, veliparib, and fuzuloparib, have shown encouraging results in the treatment of epithelial ovarian cancer (high-grade serous or endometrioid adenocarcinoma) in phase III clinical trials in recent years [4,5,6,7,8,9,10]. However, the survival benefit of these PARP inhibitors is primarily seen in patients with HRD, including *BRCA* mutations. Although there is some variation across studies and agents, there is no or small survival benefit in patients without HRD compared to patients with HRD.

Approximately 15% of non-mucinous epithelial ovarian cancer patients carry germline mutations in *BRCA1* or *BRCA2*, and approximately 5–7% of ovarian cancer patients have somatic *BRCA* mutations. This means that approximately 20–25% of patients with epithelial ovarian cancer based on *BRCA* mutation may be candidates for PARP inhibitors [11]. One study revealed that 31 percent of ovarian carcinomas, including serous, endometrioid, clear cell, and carcinosarcoma histologic types, harbored mutations in one or more of the 13 homologous recombination genes [12]. When considering epithelial ovarian cancer patients with HRD as candidates for PARP inhibitor application, approximately 30% would qualify as eligible recipients. This, in turn, implies that in around 70% of patients, the PARP inhibitor may not be effective or may result in unsatisfactory outcomes. If there is a way to overcome PARP inhibitor resistance in ovarian cancer patients without HRD, it could potentially extend the benefits of PARP inhibitors to a larger number of patients.

Trabectedin (Yondelis), formerly known as ET-743 during its development, is an antitumor agent initially discovered in the Caribbean marine tunicate, Ecteinascidia turbinata, and is now produced synthetically. Trabectedin is currently used for the treatment of patients with soft tissue sarcomas. Trabectedin is known for its unique mechanism of action, binding to the DNA minor groove to disrupt genetic transcription and inhibit cell division. Additionally, it interferes with the nucleotide-excision repair (NER) system, inducing single-strand DNA breaks. Several preclinical studies have demonstrated a synergistic effect when combining PARP inhibitors and trabectedin in sarcomas [13,14,15] or breast cancer [16]. In a study involving breast cancer cell lines, synergistic effects were observed, regardless of the presence of *BRCA* mutations [16]. These suggest the potential of trabectedin combination therapy as a promising approach to overcome PARP inhibitor resistance in ovarian cancer without *BRCA* mutations.

Based on this background, we conducted this study on the synergic effects of the combination therapy of niraparib and trabectedin using epithelial ovarian cancer cell lines without *BRCA* mutations. The researchers hypothesized that induction of BRCAness is crucial to overcoming PARP inhibitor resistance. Therefore, the main focus of this study was to investigate whether the combination therapy of niraparib and trabectedin could induce BRCAness in ovarian cancer cell lines with wild-type *BRCA* genes. Additionally, the study aimed to explore the impact on ssDNA repair enzymes such as PARP1 and PARP2 and examine whether it could induce synthetic lethality.

## 2. Results

### 2.1. Niraparib Inhibits the Proliferation of A2780 and SKOV3 Cells in a Dose- and Time-Dependent Manner

In the CCK-8 assay, A2780 and SKOV3 cells were incubated with niraparib at concentrations ranging from 0 to 50 μM for 24 h, 48 h, or 72 h. The results revealed a significant, dose- and time-dependent decrease in the proliferation of both A2780 and SKOV3 cells following niraparib treatment (Figure 1). The optimal treatment parameters for niraparib were determined by identifying the lowest concentration and treatment duration that demonstrated drug efficacy without inducing cytotoxicity, which were found to be a treatment concentration of 1 μM niraparib for 48 h.

### 2.2. Trabectedin Was Titrated to Determine the Minimum Concentration at Which It Affects Cell Viability

Subsequently, A2780 and SKOV3 cells were exposed to various concentrations of trabectedin for 48 h, and the ideal concentrations for co-administration with niraparib were determined using a CCK-8 assay. The results revealed that the optimal concentration of trabectedin was 100 pM (Figure 2).

### 2.3. Niraparib and Trabectedin Combination Therapy Demonstrated a Synergistic Effect in Cell Viability Tests, Which Was Associated with p53 Expression

Niraparib and trabectedin combination therapy exhibited a synergistic effect in cell viability tests on A2780 cells, whereas such an effect was not observed in SKOV3 cells. A2780 cells have wild-type p53, while SKOV3 cells are a p53 knock-out cell line. The difference in the synergic effect observed between the two cell lines led us to consider the potential association with this difference in p53 expression. To explore whether p53 expression is a crucial factor in the synergic effect observed in A2780 cells, we attempted to silence p53 expression using p53 siRNA. We transfected A2780 and SKOV3 cells with NC siRNA or p53 siRNA, and then treated them with trabectedin and niraparib to measure cell viability. As a result, cell viability was reduced by the two drugs in A2780 cells treated with NC siRNA, but no change was seen in A2780 cells in which p53 was silenced by p53 siRNA and in SKOV3 cells in which p53 was knocked out (Figure 3). These results confirm that the decrease in cell viability observed in A2780 cells treated with trabectedin and niraparib is dependent on p53 expression.

### 2.4. Trabectedin and Niraparib Increase p53 mRNA and Protein Levels in A2780 Cells

The mRNA expression and protein levels of p53 were assessed in both A2780 and SKOV3 cells, confirming the knockout status of p53 in SKOV3 cells. Additionally, it was observed that treatment with trabectedin and niraparib alone significantly increased the mRNA and protein levels of p53 in A2780 cells, and this increase was further enhanced when the two drugs were combined (Figure 4).

### 2.5. Trabectedin and Niraparib Lead to an Increase in Both mRNA and Protein Levels of Genes Associated with the p53-Mediated Apoptosis Pathway

As mentioned earlier, considering that the proapoptotic activity of trabectedin is primarily associated with p53, we aimed to confirm whether the effects of trabectedin and niraparib in A2780 cells are indeed mediated by p53. To do so, p53 was silenced in A2780 cells using p53 siRNA. Subsequently, the mRNA and protein levels of genes related to apoptosis were analyzed to assess the impact of p53 silencing on these genes. As a result, effective silencing of p53 in A2780 cells was confirmed. The mRNA and protein levels of genes associated with the p53-mediated apoptosis pathway (PUMA, NOXA, BAX, BAK) increased upon treatment with trabectedin and niraparib, with a further enhancement when both drugs were administered together. Additionally, the impact on p21, involved in cell cycle arrest, was more pronounced. However, in A2780 cells with p53 silencing, the effects of both drugs were no longer observed (Figure 5).

### 2.6. Trabectedin and Niraparib Reduce the mRNA and Protein Levels of SSB Repair-Related Genes in A2780 Cells

The mRNA and protein levels of genes related to single-strand break (SSB) repair were measured under the same conditions as in the previous experiment. In A2780 cells, treatment with trabectedin and niraparib resulted in decreased mRNA and protein levels of PARP1 and PARP2, with this effect being further amplified when the two drugs were used together. However, when p53 was silenced in A2780 cells, these effects disappeared. On the other hand, in SKOV3 cells, no significant changes were observed in both mRNA and protein levels of PARP1 and PARP2 (Figure 6). These results suggest that the reduction in PARP1 and PARP2 levels caused by trabectedin and niraparib in A2780 cells is dependent on p53. Western blot analyses in Figure 6B were performed under the same experimental conditions and on the same day as those shown in Figure 4 and Figure 5. Since β-actin loading controls were already validated in Figure 4 and Figure 5, and protein quantification data are provided in Figure 6C, and an additional β-actin blot was not included in Figure 6B.

### 2.7. Trabectedin and Niraparib Reduce the mRNA and Protein Levels of DSB Repair-Related Genes in A2780 Cells, and Their Combination Increases DNA Damage

Next, mRNA and protein levels of genes related to double-strand break (DSB) repair were measured. Similar to the previous experimental results, the mRNA and protein levels of BRCA1, BRCA2, and RAD51 decreased when treated with trabectedin and niraparib, with a further decrease observed when the two drugs were used together. However, no change was observed in A2780 cells in which p53 was silenced or in SKOV3 cells in which p53 was knocked out. Additionally, the DNA damage marker γ-H2AX significantly increased when the two drugs were used together (Figure 7). These results suggest that trabectedin and niraparib act synergistically to increase DNA damage by disrupting both single-strand break (SSB) and DSB repair pathways, and this effect is dependent on p53.

### 2.8. The Synthetic Lethality Effect Induced by the Combination of Trabectedin and Niraparib Is Mediated Through the p53-Dependent Apoptosis Pathway

To further confirm the p53-dependent apoptosis caused by trabectedin and niraparib, A2780 and SKOV3 cells were stained with Annexin V-FITC and PI, and then analyzed using flow cytometry (FCM). In comparison to the control (2.4% Annexin V+/PI−, indicating early apoptosis, and 2.7% Annexin V+/PI+, indicating late apoptosis) in NC siRNA-transfected A2780 cells, treatment with trabectedin alone (2.0% Annexin V+/PI− and 2.9% Annexin V+/PI+) did not significantly increase apoptosis, while treatment with niraparib alone (5.9% Annexin V+/PI− and 3.8% Annexin V+/PI+) showed a slight increase. However, when trabectedin and niraparib were administered together (40.0% Annexin V+/PI− and 15.4% Annexin V+/PI+), apoptosis was markedly increased (Figure 8). These changes were not observed in p53 siRNA-transfected A2780 cells and p53 knockout SKOV3 cells. These results suggest that the synthetic lethality effect induced by the combination of trabectedin and niraparib is mediated through the p53-dependent apoptosis pathway.

## 3. Discussion

Epithelial ovarian cancer is diagnosed in 70–80% of patients at advanced stages (Stage 3, 4), and even after undergoing debulking surgery and platinum-taxane combination chemotherapy, it continues to exhibit a poor prognosis with high rates of recurrence and mortality. In reviewing the results of phase III clinical trials reported in recent years, one of the most noteworthy advancements in the treatment of epithelial ovarian cancer is the emergence of PARP inhibitors. Olaparib demonstrated an improvement in progression-free survival in patients with platinum-sensitive, relapsed ovarian cancer and *BRCA1/2* mutations [17]. Olaparib showed a progression-free survival benefit in maintenance therapy even after primary treatment in newly diagnosed ovarian cancer patients. Through a 7-year follow-up, these improvements in progression-free survival were demonstrated to lead to an overall survival improvement [4,18]. Successful phase III clinical trial results have been sequentially reported for other PARP inhibitors such as niraparib, rucaparib, veliparib, and fuzuloparib [6,8,9,10,19]. The efficacy of the combination therapy of olaparib and bevacizumab is also noteworthy [20].

The PARP enzyme plays a crucial role in repairing single-strand DNA breaks, and PARP inhibitors impede this repair process by suppressing the enzyme’s activity. In cases of Homologous Recombination Deficiency (HRD), such as those with *BRCA* mutations, there is a difficulty in repairing double-strand DNA breaks through homologous recombination. Consequently, DNA repair predominantly relies on the single-strand DNA repair system using PARP enzyme. Administering PARP inhibitors to patients with HRD results in engaging in error-prone double-strand DNA repair mechanisms like Non-Homologous End Joining (NHEJ). This error-prone DNA repair leads to the accumulation of mutations, inducing synthetic lethality in cancer cells [21,22].

Considering such an action mechanism, theoretically, PARP inhibitors are expected to be effective when Homologous Recombination Deficiency (HRD) is present. However, clinical research results suggest that effectiveness may be observed irrespective of the HRD status. Niraparib, rucaparib, and fuzuloparib demonstrated DFS benefits regardless of the presence of HRD or *BRCA* mutations [6,10,19]. However, upon a more detailed examination of those research findings, the efficacy of PARP inhibitors is less evident in patients without *BRCA* mutations or HRD, compared to those with *BRCA* mutations or Homologous Recombination Deficiency (HRD). The likely reason for these results may be attributed to the imperfect nature of current HRD tests, raising the possibility of HRD even in cases where the test indicates HRD negativity.

In 20–25% of patients with non-mucinous epithelial ovarian cancer, there are germline or somatic *BRCA* mutations [11], and approximately 30% of patients have one or more mutations in homologous recombination genes [12]. When considering the patient group expected to respond to PARP inhibitors with the presence of HRD, it is estimated that approximately 70% of epithelial ovarian cancer patients have PARP inhibitor resistance. Expanding the application scope of PARP inhibitors to patients without HRD could be a strategy that contributes to the improved therapeutic outcomes in ovarian cancer. As one of such strategies, we hypothesized that if there were drugs capable of inducing BRCAness in ovarian cancer cells without HRD, combining them with PARP inhibitors could potentially induce synthetic lethality.

Trabectedin (Yondelis^®^, ecteinascidin-743, ET-743) is a natural marine compound with antitumor activity approved for the treatment of unresectable or metastatic soft tissue sarcoma, specifically liposarcoma or leiomyosarcoma. It is used in combination with doxorubicin for patients with uterine leiomyosarcoma [23]. The structure of trabectedin results in two main effects: the formation of a covalent bond with DNA at the N2-guanine of the minor groove, allowing trabectedin to interact with DNA, and the ability to protrude from the DNA helix, making it accessible to bind with DNA-binding molecules, such as transcriptional factors and DNA repair proteins. Another significant effect of the linkage between trabectedin and the DNA groove is that a segment of the helix becomes recognizable by the nucleotide excision repair (NER) system, resulting in the accumulation of DNA–trabectedin–protein repair complexes. As a result, the formation of DSBs, cell cycle arrest in G2-M phase, and induction of p53-independent apoptosis are primary effects of this phenomenon. Previous in vitro investigations have indicated that trabectedin effectiveness is influenced by the status of both NER and HR DNA repair pathways [24,25,26,27]. Trabectedin has been observed to be more effective against cells with HR deficiency compared to normal cells [28]. Although trabectedin’s action mechanism has not been fully elucidated, it is known not to directly cause dsDNA breaks [29].

Combined therapy with PARP inhibitors and trabectedin has been reported to demonstrate a synergistic effect in several preclinical studies conducted on sarcoma and breast cancer models [13,14,15,16]. In a study using breast cancer cell lines [16], the combination of trabectedin and olaparib induces an artificial synthetic lethality effect, regardless of *BRCA1* status. The study showed that the combination treatment was associated with a strong accumulation of double-stranded DNA breaks, G2/M arrest, and apoptotic cell death. However, synergistic effects were not observed when trabectedin was combined with veliparib or iniparib [16].

The primary mechanism known for trabectedin to date is the inhibition of the nucleotide excision repair (NER) system. Therefore, when used in conjunction with PARP inhibitors, it results in dual inhibition of systems involved in ssDNA break repair (inhibition of NER by trabectedin and inhibition of PARP by olaparib). Unrepaired ssDNA breaks can progress to dsDNA breaks, leading to the accumulation of dsDNA breaks. In such a scenario, it would be predicted that BRCA-proficient cancer cells need to enhance the activation of homologous recombination factors such as BRCA1, BRCA2, and RAD51 to repair dsDNA breaks.

However, the results of this study did not support this prediction. In the findings of this study, not only were PARP1 and PARP2, involved in ssDNA break repair, more inhibited in the combination therapy of trabectedin and niraparib than in either agent alone, but factors associated with dsDNA break homologous recombination repair (BRCA1, BRCA2, and RAD51) were also suppressed. This was accompanied by an increase in γ-H2AX, indicating DNA damage, along with a decrease in cell viability, ultimately leading to apoptosis. These results were observed in A2780 cells (with wild-type p53), but not in SKOV3 cells (p53 knockout). Additionally, the synergistic effect observed in A2780 was found to be attenuated upon the loss of p53 through p53 siRNA. This study demonstrated that the combined therapy of trabectedin and PARP inhibitors can induce BRCAness and elicit a synthetic lethality effect in BRCA-proficient epithelial ovarian cancer cells. It also highlighted the significance of the p53-mediated apoptotic pathway in such a synergistic effect. This result suggests that there may be an as-yet-unknown mechanism by which trabectedin, in combination with niraparib, induces homologous recombination deficiency.

There are likely to be numerous hurdles in demonstrating the effectiveness of the combined therapy in HR-proficient epithelial ovarian cancer patients. One of them is because *TP53* mutations are commonly found in most epithelial ovarian cancers. The mutations are predominantly in the order of missense, frameshift, and nonsense, with the majority exhibiting abnormalities in p53 function [30]. Therefore, these research findings emphasize the importance of p53 function recovery therapy in the treatment of epithelial ovarian cancer. Research on the combination therapy with p53 function recovery agents such as PRIMA-1, PRIMA-1Met [31], and ReACp53 [32] is considered necessary.

The SKOV3 cell line, with its p53 gene knockout, cannot be considered representative of epithelial ovarian cancer, given that the majority of *TP53* mutations in this type of cancer are missense mutations. Therefore, additional research using a broader range of epithelial ovarian cancer cell lines is necessary, and the efficacy needs to be verified through animal experiments using xenografts. Furthermore, there is a need for research on the effectiveness of trabectedin in combination with other PARP inhibitors, apart from niraparib.

For actual clinical use as a treatment, a survival benefit must be demonstrated through clinical trials. However, there is currently a scarcity of clinical trials investigating the combined use of trabectedin and PARP inhibitors. There was a phase 1b study using trabectedin and olaparib in patients with advanced and non-resectable bone and soft-tissue sarcomas. If future clinical trials are conducted, the results of this study may serve as a reference for drug dosage determination [33].

A limitation of our study is that we did not directly assess the involvement of ATM and ATR pathways in the observed DNA damage response. Given that niraparib is a PARP inhibitor that induces replication stress and trabectedin interferes with transcription-coupled repair, it is plausible that both ATR and ATM signaling may contribute to the accumulation of DNA damage. Future studies should investigate the activation status of these pathways using phosphorylation markers (e.g., p-ATR, p-ATM, p-CHK1, p-CHK2) and pharmacological inhibitors to further elucidate the molecular mechanisms underlying the synergistic effects observed in this study.

In conclusion, the combination therapy of niraparib and trabectedin inhibits PARP1 and PARP2 involved in ssDNA break repair, and suppresses BRCA1, BRCA2, and RAD51 involved in dsDNA break homologous recombination. This leads to an increase in DNA damage, inhibiting cell viability and enhancing p53-mediated apoptosis. Trabectedin, when used in combination with PARP inhibitors, could be a strategy to induce BRCAness and trigger synthetic lethality, particularly in HRD-proficient ovarian cancer.

## 4. Materials and Methods

### 4.1. Cell Culture and Reagents

The ovarian cancer cell lines A2780 and SKOV3, derived from epithelial ovarian carcinoma of an untreated patient were provided by Dr. IM Shih (Johns Hopkins School of Medicine, Baltimore, MD, USA). The cells were maintained in RPMI 1640 media (Cytiva, Marlborough, MA, USA) supplemented with 5% FBS (Gibco, Thermo Fisher Scientific, Waltham, MA, USA) and 1% penicillin-streptomycin antibiotics in an incubator with 5% CO_2_ at 37 °C. The details of the cell culture conditions are summarized in Table 1. Trabectedin and niraparib were obtained from MedChemExpress (MedChemExoress, Monmouth Junction, NJ, USA) and dissolved in dimethyl sulfoxide (DMSO).

### 4.2. Small Interfering RNA (siRNA) Transfection

Negative control (NC) siRNAs (10 nM) and p53 siRNAs (10 nM) were transfected into A2780 and SKOV3 cells grown on 6-well plates (SPL, Pocheon, Republic of Korea) with Lipofectamine RNAiMAX according to the manufacturer’s protocol for 24 h followed by drug treatments for 48 h. The details of the gene silencing reagents used in this study are summarized in Table 2.

### 4.3. Cell Viability Assay

Logarithmically growing cells were chosen, digested with 0.25% trypsin, and then suspended in a medium supplemented with 5% FBS. Subsequently, appropriate cell concentrations were seeded in 96-well plates. After 24 h, a range of compound concentrations (with DMSO concentration no more than 0.1%) were added to the cells. Following 48 h of treatment, cells were collected, and cell viability was assessed using the CCK-8 method. The reagents used for the cell viability assay are listed in Table 3. The inhibitory activity of each compound was assessed in duplicate wells for each cell line, and the experiment was repeated three times to ensure reliability of results.

### 4.4. RNA Purification and Real-Time Quantitative PCR Analysis

Total RNA was isolated from A2780 and SKOV3 cells using Trizol Reagent (Invitrogen, Thermo Fisher Scientific, Waltham, MA, USA) and prepared for complementary DNA synthesis using the PrimeScript 1st strand cDNA Synthesis Kit (TaKaRa, Kusatsu, Shiga, Japan). The reagents used for real-time quantitative PCR are listed in Table 4. Gene expression was determined by RT-qPCR analysis using AccuPower^®^ 2X GreenStar™ qPCR Master Mix (Bioneer, Daejeon, Republic of Korea). mRNA levels were normalized using the β-Actin gene. Primer information for RT-qPCR analysis is listed in Table 5.

### 4.5. Western Blot Analysis

The cells were lysed in RIPA buffer (ELPIS, Daejeon, Republic of Korea) to extract cellular proteins. The reagents used for Western blot analysis are listed in Table 6. The resulting cell lysates were denatured and separated on a sodium dodecyl sulfate polyacrylamide gel (SDS-PAGE), followed by transfer onto nitrocellulose membranes (GE Healthcare, Chicago, IL, USA). The membranes were then blocked in 5% non-fat milk dissolved in Tris-buffered saline supplemented with Tween 20 (TBST) for 1 h at room temperature. Subsequently, they were immunoblotted with primary antibodies overnight at 4 °C. Afterward, the membranes were washed twice with TBST for 20 min each time and immunoblotted with secondary antibodies for 1 h at room temperature, followed by two washes with TBST for 20 min each. Protein bands were detected using SuperSignal™ West Femto Maximum Sensitivity Substrate (Thermo Scientific, Waltham, MA, USA) in a ChemiDoc™ XRS+ System (Bio-Rad, Hercules, CA, USA), with β-actin (Cell Signaling Technology, Danvers, MA, USA) used as the loading control. The antibodies used for Western blot analysis are summarized in Table 7. The bands were quantified using ImageJ software.

### 4.6. Annexin V/PI Assay

A2780 and SKOV3 cells were seeded in 6-well plates (SPL, 30006) and allowed to attach. After 24 h of initial culture, the cells were transfected with NC siRNA or p53 siRNA and were cultured for an additional 24 h. Subsequently, control (DMSO), trabectedin, niraparib, or trabectedin + niraparib were added to the cells and incubated for 48 h, respectively. The reagents used for the Annexin V/PI assay are listed in Table 8. Staining was performed in triplicate in two separate experiments using Annexin V-fluorescein isothiocyanate (FITC) and propidium iodide (PI) and analyzed by flow cytometry (BD Biosciences, Franklin Lakes, NJ, USA, BD FACSAria III), according to the manufacturer’s protocol (Merck, APOAF-20TST, Darmstadt, Germany).

### 4.7. Statistics

The numbers of ’N’ for each group used in the experiments are indicated in the figure legends. Statistical analyses were performed using the two-tailed independent *t*-test, and error bars represent means ± s.e.m. (standard error of the mean). A *p*-value of <0.05 was considered a statistically significant difference.

## 5. Conclusions

Niraparib and trabectedin synergistically induce synthetic lethality in BRCA-proficient ovarian cancer cells. The therapy inhibits ssDNA and dsDNA repair pathways, triggering p53-dependent apoptosis. Synergistic effects are observed in A2780 cells and are lost with p53 silencing. This highlights p53’s critical role in mediating the therapy’s effects.

## Figures and Tables

**Figure 1 ijms-26-02921-f001:**
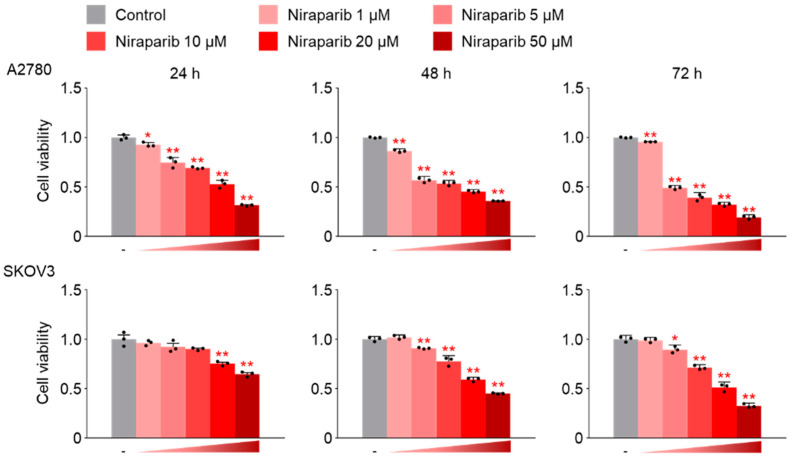
Niraparib effectively inhibits the proliferation of A2780 and SKOV3 cells. Cell viability was assessed using the CCK-8 assay to determine the optimal dose and treatment time for niraparib (a PARP inhibitor). A2780 and SKOV3 cells were cultured and treated with varying concentrations of niraparib for 24, 48, and 72 h, followed by the CCK-8 assay. The normalized values of the control group were set as fold 1 (* *p* < 0.05, ** *p* < 0.01 vs. Control; group/n = 5). Data represent mean ± s.e.m. Statistical analysis was performed using a two-tailed *t*-test. The black dots represent the measurement values from three separate experiments.

**Figure 2 ijms-26-02921-f002:**
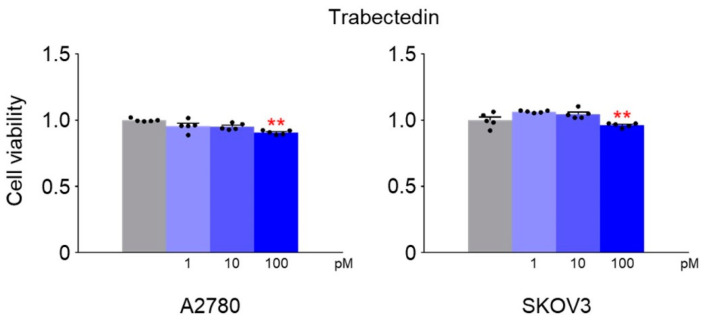
Cell viability changes in A2780 and SKOV3 cells by trabectedin. A2780 and SKOV3 cells were cultured and treated with varying concentrations of trabectedin for 48 h, followed by CCK-8 assay. The normalized values of the control group were set as fold 1 (** *p* < 0.01 vs. Control; group/n = 5). Data represent mean ± s.e.m. Statistics by two-tailed *t*-test.

**Figure 3 ijms-26-02921-f003:**
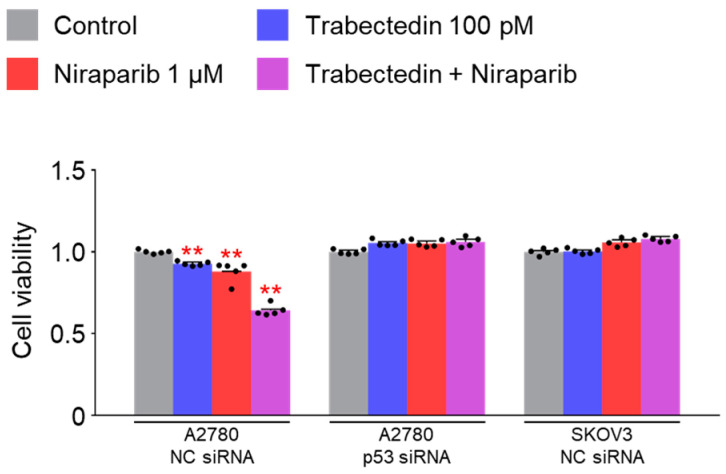
Cell viability decreased in A2780 cells by trabectedin and niraparib is dependent on p53. Cell viability of A2780 and SKOV3 cells was measured by CCK-8 assay after transfection with NC siRNA or p53 siRNA for 24 h, treatment with trabectedin, niraparib, or trabectedin + niraparib for 48 h. The normalized values of the control group were set as fold 1 (** *p* < 0.01 vs. each Control; group/n = 5). Data represent mean ± s.e.m. Statistics by two-tailed *t*-test.

**Figure 4 ijms-26-02921-f004:**
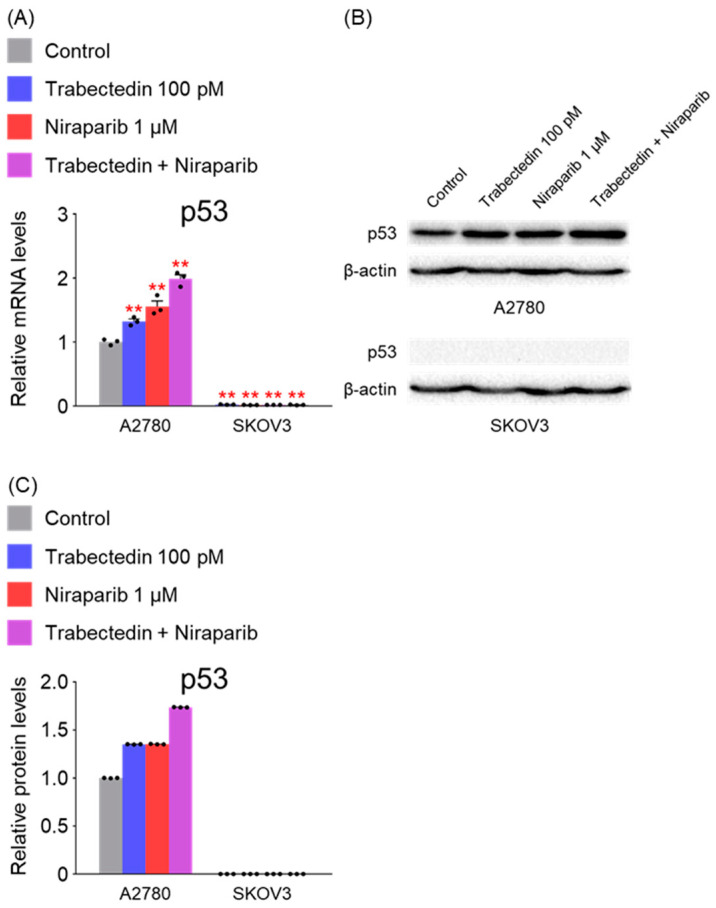
Trabectedin and niraparib increased p53 mRNA and protein levels in A2780 cells. (**A**) The expression levels of p53 in A2780 and SKOV3 cells were determined 48 h after treatment with trabectedin, niraparib, or a combination of trabectedin and niraparib using RT-qPCR analysis. mRNA levels were normalized to β-actin, with the normalized values of the A2780 control group set as fold 1 (** *p* < 0.01 vs. Control; group/n = 3). Data represent mean ± s.e.m., and statistical analysis was performed using a two-tailed *t*-test. (**B**) Cell lysates from A2780 and SKOV3 cells treated with trabectedin (100 pM), niraparib (1 μM), or a combination of trabectedin and niraparib (100 pM/1 μM) for 48 h were analyzed, with β-actin used as a loading control. (**C**) Quantification of each band was performed using ImageJ software version 1.54b.

**Figure 5 ijms-26-02921-f005:**
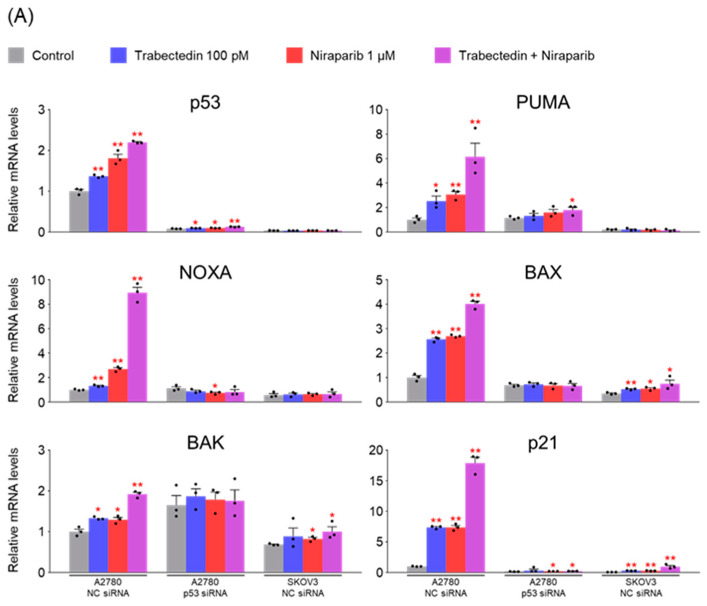
Trabectedin and niraparib lead to an increase in both mRNA and protein levels of genes associated with the p53-mediated apoptosis pathway. (**A**) The expression levels of genes related to the p53-mediated apoptosis pathway in A2780 and SKOV3 cells were measured by RT-qPCR after treatment with NC siRNA or p53 siRNA for 24 h, followed by treatment with trabectedin, niraparib, or trabectedin + niraparib for 48 h. mRNA levels are relative to β-actin. The normalized values of the A2780 NC siRNA control group were set as fold 1 (* *p* < 0.05, ** *p* < 0.01 vs. Control; group/n = 3). Data represent mean ± s.e.m. Statistics by two-tailed *t*-test. (**B**) The lysates from A2780 and SKOV3 cells were transfected with either NC siRNA or p53 siRNA for 24 h, and then treated with trabectedin, niraparib, or trabectedin + niraparib for 48 h. β-actin was used as a loading control. (**C**) Each band was quantified using ImageJ software.

**Figure 6 ijms-26-02921-f006:**
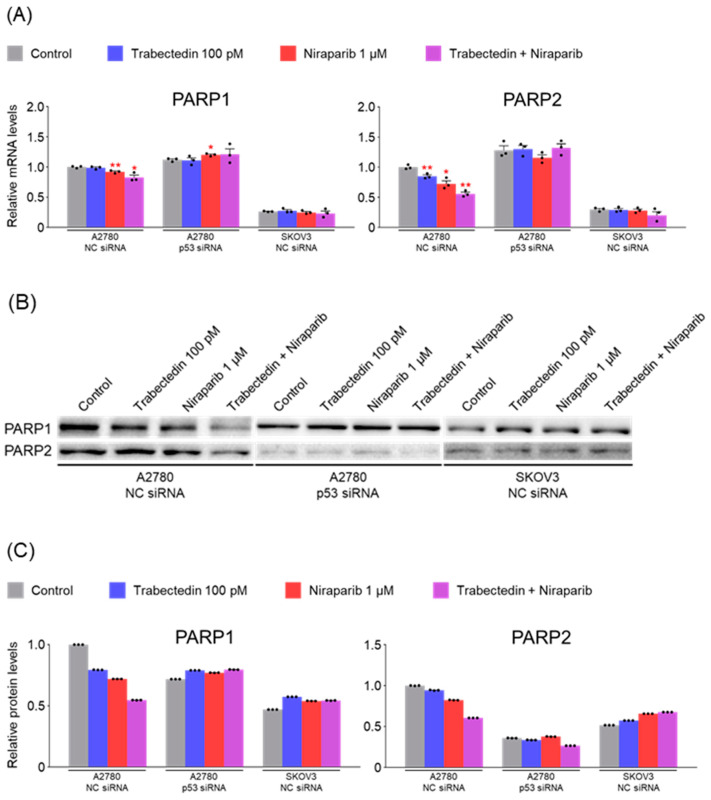
Trabectedin and niraparib reduce the mRNA and protein levels of SSB repair-related genes in A2780 cells. (**A**) The expression levels of PARP1 and PARP2 in A2780 and SKOV3 cells were measured by RT-qPCR analysis after treatment with either NC siRNA or p53 siRNA for 24 h, followed by treatment with trabectedin, niraparib, or a combination of trabectedin and niraparib for 48 h. mRNA levels were normalized to β-actin, with the normalized values of the A2780 NC siRNA control group set as fold 1 (* *p* < 0.05, ** *p* < 0.01 vs. Control; group/n = 3). Data represent mean ± s.e.m., and statistical analysis was performed using a two-tailed *t*-test. (**B**) Cell lysates from A2780 and SKOV3 cells were transfected with either NC siRNA or p53 siRNA for 24 h, followed by treatment with trabectedin, niraparib, or a combination of trabectedin and niraparib for 48 h. (**C**) Quantification of each band was performed using ImageJ software.

**Figure 7 ijms-26-02921-f007:**
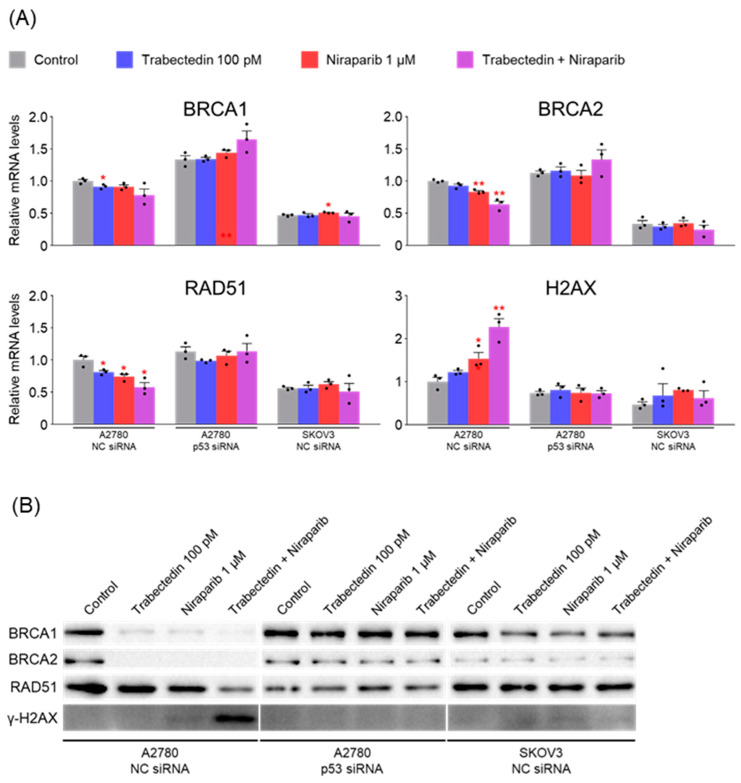
Trabectedin and niraparib reduce the mRNA and protein levels of DSB repair-related genes in A2780 cells, and their combination increases DNA damage. (**A**) The expression levels of genes related to double-strand break (DSB) repair in A2780 and SKOV3 cells were assessed by RT-qPCR analysis after treatment with either NC siRNA or p53 siRNA for 24 h, followed by treatment with trabectedin, niraparib, or a combination of trabectedin and niraparib for 48 h. mRNA levels were normalized to β-actin, with the normalized values of the A2780 NC siRNA control group set as fold 1 (* *p* < 0.05, ** *p* < 0.01 vs. Control; group/n = 3). Data represent mean ± s.e.m., and statistical analysis was performed using a two-tailed *t*-test. (**B**) Cell lysates from A2780 and SKOV3 cells were transfected with either NC siRNA or p53 siRNA for 24 h, followed by treatment with trabectedin, niraparib, or a combination of trabectedin and niraparib for 48 h. (**C**) Quantification of each band was performed using ImageJ software.

**Figure 8 ijms-26-02921-f008:**
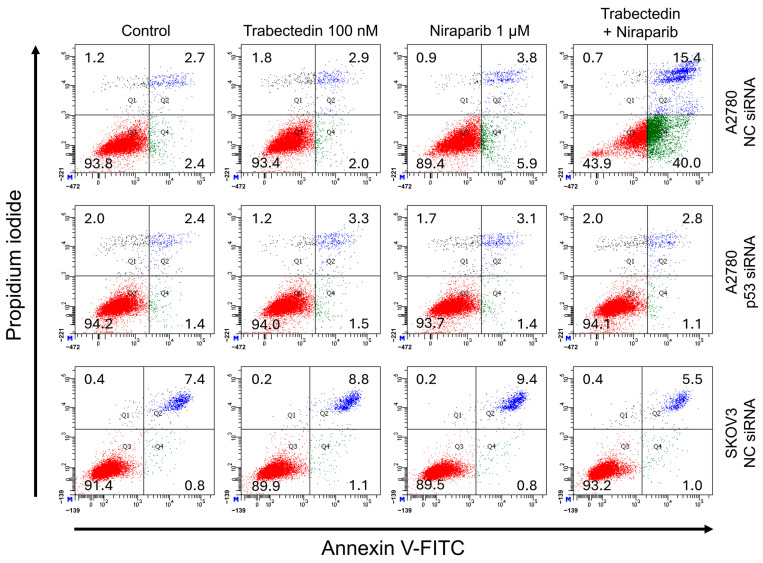
Identification of apoptosis through FCM assay using Annexin V/PI double staining. FCM assay is represented by dot plot diagrams that demonstrate the typical apoptotic cell population with Annexin V-FITC and PI staining. The upper left quadrants (Q1) of the panels show necrotic cells, which were negative for Annexin V and positive for PI; the upper right quadrants (Q2) represent late apoptotic cells, which were positive for both Annexin V and PI; the lower left quadrants (Q3) represent the intact viable cells, which were negative for both Annexin V and PI; the lower right quadrants (Q4) represent early apoptotic cells, which were positive for Annexin V and negative for PI.

**Table 1 ijms-26-02921-t001:** Cell culture reagents.

Cell Lines and Cell Culture		
Reagent or Resource	Source	Identifier
**Chemicals and Reagents**		
HyClone RPMI 1640 media	Cytiva	Cat# SH30027.01
Fetal bovine serum (FBS)	Gibco	Cat# 26140079
Penicillin-Streptomycin (10,000 U/mL)	Gibco	Cat# 15140122
0.25 trypsin-edta (1x)	Gibco	Cat# 25200056
Phosphate-Buffered Saline (1X)	Corning, Corning, NY, USA	Cat# 21-040-CV
Niraparib	MedChemExpress	Cat# HY-10619
Trabectedin	MedChemExpress	Cat# HY-50936
**Experimental Models: Cell line**		
Cell: A2780		
Cell: SKOV3		

**Table 2 ijms-26-02921-t002:** Gene silencing reagents.

Gene Silencing		
Reagent or Resource	Source	Identifier
**Chemicals and Reagents**		
Lipofectamine RNAiMAX Transfection Reagent	Invitrogen	Cat# 13778150
Opti-MEM^®^ I Reduced Serum Medium	Gibco	Cat# 31985-070
p53 siRNA AccuTarget^™^ Genome-wide Predesigned siRNA	Bioneer	Cat# 7157-1
AccuTarget^™^ Negative Control siRNA	Bioneer	

**Table 3 ijms-26-02921-t003:** Cell viability assay reagents.

Cell Viability Assay		
Reagent or Resource	Source	Identifier
**Chemicals and Reagents**		
Cell Counting Kit-8	Dojindo, Kumamoto, Japan	Cat# CK04-11
96 Well Cell Culture Plates	SPL	Cat# 30096
**Machines and Software**		
SpectraMax^®^ iD5 Multi-Mode Microplate Reader	Moleculardevices, San Jose, CA, USA	
SoftMax Pro Software, Version 7.1	Moleculardevices	

**Table 4 ijms-26-02921-t004:** Real-time quantitative PCR reagents.

Real-Time Quantitative PCR		
Reagent or Resource	Source	Identifier
**Chemicals and Reagents**		
PrimeScript^™^ 1st strand cDNA Synthesis Kit	TaKaRa	Cat# 6110A
AccuPower^®^ 2X GreenStar^™^ qPCR Master Mix	Bioneer	Cat# K-6254
Trizol	Invitrogen	Cat# 15596018
**Machines and Software**		
CFX Opus 96 Real-Time PCR System	Bio-Rad	Cat# 12011319
T100 Thermal Cycler, Version 1.201	Bio-Rad	Cat# 1861096

**Table 5 ijms-26-02921-t005:** Primers for real-time quantitative PCR analysis.

Genes		Sequence
*p53*	Forward	CCTCAGCATCTTATCCGAGTGG
	Reverse	TGGATGGTGGTACAGTCAGAGC
*p53* (silencing)	Forward	CCCCTCTGAGTCAGGAAACAT
	Reverse	TTCATCTGGACCTGGGTCTTC
*PUMA*	Forward	ACGACCTCAACGCACAGTACGA
	Reverse	CCTAATTGGGCTCCATCTCGGG
*NOXA*	Forward	CTGGAAGTCGAGTGTGCTACTC
	Reverse	TGAAGGAGTCCCCTCATGCAAG
*BAX*	Forward	TCAGGATGCGTCCACCAAGAAG
	Reverse	TGTGTCCACGGCGGCAATCATC
*BAK*	Forward	TTACCGCCATCAGCAGGAACAG
	Reverse	GGAACTCTGAGTCATAGCGTCG
*p21*	Forward	AGGTGGACCTGGAGACTCTCAG
	Reverse	TCCTCTTGGAGAAGATCAGCCG
*PARP1*	Forward	CCAAGCCAGTTCAGGACCTCAT
	Reverse	GGATCTGCCTTTTGCTCAGCTTC
*PARP2*	Forward	GGTGGCTTGTTCAGGCAATCTC
	Reverse	GGTGGCATAGTCCATCTGTAGC
*BRCA1*	Forward	CTGAAGACTGCTCAGGGCTATC
	Reverse	AGGGTAGCTGTTAGAAGGCTGG
*BRCA2*	Forward	GGCTTCAAAAAGCACTCCAGATG
	Reverse	GGATTCTGTATCTCTTGACGTTCC
*RAD51*	Forward	TCTCTGGCAGTGATGTCCTGGA
	Reverse	TAAAGGGCGGTGGCACTGTCTA
*H2AX*	Forward	CGGCAGTGCTGGAGTACCTCA
	Reverse	AGCTCCTCGTCGTTGCGGATG
*β-Actin*	Forward	CACCATTGGCAATGAGCGGTTC
	Reverse	AGGTCTTTGCGGATGTCCACGT

**Table 6 ijms-26-02921-t006:** Western blot analysis reagents.

Western Blot Analysis		
Reagent or Resource	Source	Identifier
**Chemicals and Reagents**		
30% Acrylamide/Bis Solution, 29:1	Bio-Rad	Cat# 1610156
Tetramethylethylenediamine (TEMED)	Amresco, Solon, OH, USA	Cat# M146-25 mL
Ammonium persulfate (APS)	Amresco	Cat# M133-25G
10X TBS with Tween20	Biosesang, Seongnam, Republic of Korea	Cat# TR2007-100
10X PBS pH7.4	Biosesang	Cat# PR4007-100
1.5 M Tris Buffer, pH 8.8	GenDepot, Barker, TX, USA	Cat# T8101-050
0.5 M Tris Buffer, pH 6.8	GenDEPOT	Cat# T8102-050
Tris	Amresco	Cat# 0497-1 kg
Glycine	Bio-Rad	Cat# 1610718
SODIUM DODECYL SULFATE (SDS)	Biosesang	Cat# SR1010-500
Precision Plus Protein^™^ Dual Color Standards	Bio-Rad	Cat# 1610374
Amersham Protran Premium 0.45 NC	GE Healthcare	Cat# 10600003
NuPAGE^™^ LDS Sample Buffer (4X)	Invitrogen	Cat# NP0007
Pierce^™^ ECL Western Blotting Substrate	Thermo Scientific	Cat# 32106
SuperSignal^™^ West Femto Maximum Sensitivity Substrate	Thermo Scientific	Cat# 34095
Protein Extraction Solution (RIPA)	ELPIS	Cat# EBA-1149
Methanol	MercK, Darmstadt, Germany	Cat# 1.06009.1011
Reagent or Resource	Source	Identifier
Mini Trans-Blot Electrophoretic Transfer Cell	Bio-Rad	Cat# BR1703930
Pierce BCA Protein Assay Kit	Thermo Scientific	Cat# 23227
**Machines and Software**		
ChemiDoc^™^ XRS+ System	Bio-Rad	Cat# 1708265
Image Lab^™^ Software	Bio-Rad	
ImageJ	National Institutes of Health, Bethesda, MD, USA	

**Table 7 ijms-26-02921-t007:** Antibodies for Western blot analysis.

Antibodies		
Reagent or Resource	Source	Identifier
p53 Rabbit Ab	Cell Signaling	Cat# 9282S
BRCA1 Rabbit mAb	Cell Signaling	Cat# 9010S
BRCA2 (D9S6V) Rabbit mAb	Cell Signaling	Cat# 10741S
Anti-PARP-1 Antibody (B-10)	Santa Cruz Biotechnology, Dallas, TX, USA	Cat# sc-74470
Anti-PARP-2 Antibody (F-3)	Santa Cruz Biotechnology	Cat# sc-393310
RAD51 (D4B10) Rabbit mAb	Cell Signaling	Cat# 8875S
PUMA (E2P7G) Rabbit mAb	Cell Signaling	Cat# 98672S
NOXA (D8L7U) Rabbit mAb	Cell Signaling	Cat# 14766S
p21 Waf1/Cip1 (12D1) Rabbit mAb	Cell Signaling	Cat# 2947S
BAX (E4U1V) Rabbit mAb	Cell Signaling	Cat# 41162S
BAK (D4E4) Rabbit mAb	Cell Signaling	Cat# 12105S
β-Actin (8H10D10) Mouse mAb	Cell Signaling	Cat# 3700S
Anti-mouse IgG, HRP-linked Antibody	Cell Signaling	Cat# 7076S
Anti-rabbit IgG, HRP-linked Antibody	Cell Signaling	Cat# 7074S

**Table 8 ijms-26-02921-t008:** Annexin V/PI assay reagents.

Annexin V/PI Assay		
Reagent or Resource	Source	Identifier
**Chemicals and Reagents**		
Annexin V-FITC Apoptosis Detection Kit	Merck	Cat# APOAF-20TST
**Machines and Software**		
BD FACSAria III	BD Biosciences	
BD FACSDiva™ Software, Version 9.0	BD Biosciences	

## Data Availability

Data are available from the corresponding authors upon reasonable request.

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
