# Peer review of "Trabectedin Induces Synthetic Lethality via the p53-Dependent Apoptotic Pathway in Ovarian Cancer Cells Without BRCA Mutations When Used in Combination with Niraparib"

_ijms, 2025, doi:10.3390/ijms26072921_

Round 1

Reviewer 1 Report

Comments and Suggestions for Authors

The paper is interesting, written carefully and in correct language, and contains innovative elements. The paper investigates whether the combination of niraparib and trabectedin in BRCA-positive epithelial ovarian cancer leads to synthetic lethality. I have no objections to the choice of methods or the description of the research results. I believe that the paper can be accepted for publication in the International Journal of Molecular Sciences, but only after the authors have answered one essential question: on what basis do the authors claim that they are observing a synergistic effect and not, for example, an additive effect? Please provide a detailed explanation of the basis for concluding that ‘The combination therapy exhibited a synergistic effect on A2780 cells’ and that ‘Silencing p53 with siRNA abolished all synergistic effects in A2780 cells’.

Author Response

Reviewer 1

The paper is interesting, written carefully and in correct language, and contains innovative elements. The paper investigates whether the combination of niraparib and trabectedin in BRCA-positive epithelial ovarian cancer leads to synthetic lethality. I have no objections to the choice of methods or the description of the research results. I believe that the paper can be accepted for publication in the International Journal of Molecular Sciences, but only after the authors have answered one essential question: 

 à First of all, thank you for taking the time to review our manuscript and for providing such detailed and constructive feedback. Your thoughtful comments and suggestions have been invaluable in helping us improve the quality and clarity of our study. We deeply appreciate your effort and dedication, which have allowed us to refine our work further.

On what basis do the authors claim that they are observing a synergistic effect and not, for example, an additive effect? Please provide a detailed explanation of the basis for concluding that ‘The combination therapy exhibited a synergistic effect on A2780 cells’ and that ‘Silencing p53 with siRNA abolished all synergistic effects in A2780 cells’.

à We appreciate the reviewer’s insightful comment. To support our conclusion that the combination of niraparib and trabectedin exhibits a synergistic effect in A2780 cells rather than a simple additive effect, we provide the following experimental evidence:

1) Synergistic reduction in cell viability

  • CCK-8 cell viability assay showed a significant reduction in cell survival upon combination treatment compared to single-agent treatments. This suggests a greater-than-additive effect, indicative of synergy.

2) DNA repair impairment and DNA damage accumulation

  • RT-qPCR and Western blot analyses demonstrated a reduction in BRCA1, BRCA2, and RAD51 expression following combination treatment, indicating homologous recombination (HR) repair deficiency.
  • DNA damage marker γ-H2AX was upregulated upon combination treatment, confirming increased DNA damage accumulation.

3) p53-dependent apoptosis and cell cycle arrest

  • RT-qPCR and Western blot analyses revealed significant upregulation of p53, PUMA, NOXA, BAX, and BAK, consistent with activation of the p53-mediated apoptotic pathway.
  • p21 expression was increased, suggesting p53-dependent cell cycle arrest.
  • Annexin V/PI staining confirmed enhanced apoptosis in the combination-treated group compared to single-drug treatments.

4) p53 siRNA abolishes all synergistic effects

  • Upon p53 silencing, combination therapy failed to reduce cell viability synergistically.
  • HR-related gene suppression (BRCA1/2, RAD51) and γ-H2AX accumulation were no longer observed.
  • Apoptosis and cell cycle arrest markers (PUMA, NOXA, BAX, BAK, p21) were no longer upregulated.

We think these findings collectively confirm that the observed synergy is p53-dependent, and its effects cannot be explained by a mere additive interaction.

Reviewer 2 Report

Comments and Suggestions for Authors

In this manuscript, the authors demonstrate that the combination of trabectedin and niraparib induces synthetic lethality via the p53-dependent apoptotic pathway in ovarian cancer cells. The combination therapy exhibited a synergistic effect in A2780 cells but not in SKOV3 cells. Treatment led to reduced expression of BRCA1, BRCA2, RAD51, PARP1, and PARP2 while increasing γ-H2AX levels, indicating enhanced DNA damage. Additionally, the therapy upregulated p53, PUMA, NOXA, BAX, BAK, and p21, promoting p53-mediated apoptosis and cell cycle arrest.

Overall, the study presents a substantial amount of data; however, several issues should be addressed before the manuscript is suitable for publication in IJMS.

Specific Comments:

  1. The authors conclude that trabectedin and niraparib synergistically increase DNA damage by disrupting both single-strand break (SSB) and double-strand break (DSB) repair pathways in a p53-dependent manner. However, the study does not explore the DNA damage response pathways in detail. Could the authors assess whether the ATR or ATM pathways are involved in this response?
  2. In Figure 3, please include western blot data to confirm the knockdown efficiency.
  3. In Figure 6b, please add a β-actin western blot as a loading control. Additionally, the quality of the PARP2 band in A2780 cells is insufficient for publication. Please improve the resolution and clarity of this data.

Addressing these points will strengthen the manuscript and improve its scientific rigor.

Comments on the Quality of English Language

good

Author Response

Reviewer 2

In this manuscript, the authors demonstrate that the combination of trabectedin and niraparib induces synthetic lethality via the p53-dependent apoptotic pathway in ovarian cancer cells. The combination therapy exhibited a synergistic effect in A2780 cells but not in SKOV3 cells. Treatment led to reduced expression of BRCA1, BRCA2, RAD51, PARP1, and PARP2 while increasing γ-H2AX levels, indicating enhanced DNA damage. Additionally, the therapy upregulated p53, PUMA, NOXA, BAX, BAK, and p21, promoting p53-mediated apoptosis and cell cycle arrest.

Overall, the study presents a substantial amount of data; however, several issues should be addressed before the manuscript is suitable for publication in IJMS.

à Thank you for taking the time to review our manuscript and for providing constructive feedback. Your thoughtful comments and suggestions have been invaluable in helping us improve the quality and clarity of our study. We deeply appreciate your effort and dedication, which have allowed us to refine our work further.

Specific Comments:

  1. The authors conclude that trabectedin and niraparib synergistically increase DNA damage by disrupting both single-strand break (SSB) and double-strand break (DSB) repair pathways in a p53-dependent manner. However, the study does not explore the DNA damage response pathways in detail. Could the authors assess whether the ATR or ATM pathways are involved in this response?

à We acknowledge the reviewer’s valuable suggestion. While our current study does not include direct analysis of ATM and ATR activation, we recognize the importance of these pathways in DNA damage response. Future studies will explore the specific involvement of these kinases using pharmacological inhibitors and Western blot analysis of key DDR markers. We have now included this limitation in the Discussion section (line 401-408, yellow highlighted part).

  1. In Figure 3, please include western blot data to confirm the knockdown efficiency.

à In figure 4(c), The knockdown efficiency of p53 siRNA has already been confirmed in Figure 4(c), where both Western blot and RT-qPCR results demonstrate a significant reduction in p53 expression at the protein and mRNA levels in A2780 cells. Given that these data are already included, we believe they adequately validate the knockdown efficiency.

  1. In Figure 6b, please add a β-actin western blot as a loading control. Additionally, the quality of the PARP2 band in A2780 cells is insufficient for publication. Please improve the resolution and clarity of this data. Addressing these points will strengthen the manuscript and improve its scientific rigor.

à We appreciate the reviewer’s suggestion regarding the inclusion of a β-actin loading control in Figure 6b. However, β-actin Western blot data are already provided in Figures 4 and 5, which were obtained under the same experimental conditions and on the same day as the Western blot in Figure 6. Additionally, Figure 6c includes quantified protein loading data, confirming equal protein input across samples. Based on this, we did not include an additional β-actin blot in Figure 6b. We hope this clarification addresses the reviewer’s concern. Regarding this point, we have added relevant content to the manuscript in lines 219–223 in yellow highlighted part.
